# Investigating Combination Therapy: The Role of Lutetium-177 PSMA-617 Radioligand Therapy and Androgen Receptor Pathway Inhibitors in Metastatic Castration-Resistant Prostate Cancer

**DOI:** 10.3390/jcm13164585

**Published:** 2024-08-06

**Authors:** Oğuzcan Kınıkoğlu, Bala Başak Öven, Serkan Çelik, Nalan Alan Selçuk, Gamze Beydağı, Kaan Akçay, Levent Kabasakal

**Affiliations:** 1Department of Medical Oncology, Health Science University, Kartal Dr. Lütfi Kirdar City Hospital, İstanbul 34865, Türkiye; 2Department of Medical Oncology, Yeditepe University Medical Faculty, İstanbul 34718, Türkiye; basakoven@yahoo.com (B.B.Ö.); mdscelik@yahoo.com (S.Ç.); 3Department of Nuclear Medicine, Yeditepe University Medical Faculty, İstanbul 34718, Türkiye; nalanalanselcuk@gmail.com (N.A.S.); gamzebeydagi@gmail.com (G.B.); kaan.akcayy97@gmail.com (K.A.); 4Department of Nuclear Medicine, Istanbul University Cerrahpaşa Medical Faculty, İstanbul 34098, Türkiye; lkabasakal@tsnm.org

**Keywords:** metastatic castration-resistant prostate cancer, Lutetium-177 PSMA-617 radioligand therapy, androgen receptor pathway inhibitors, combination therapy

## Abstract

**Background:** The combination of Lutetium-177 (Lu-177) PSMA-617 radioligand therapy (RLT) with androgen receptor pathway inhibitors (ARPIs) has shown promise in metastatic castration-resistant prostate cancer (mCRPC). However, real-world data on the efficacy and safety of this combination are limited. This study aimed to evaluate the impact of combination therapy with Lu-177 PSMA-617 RLT and ARPIs on progression-free survival (PFS) and overall survival (OS) in patients with mCRPC. **Methods:** In this retrospective study, 104 mCRPC patients receiving Lu-177 PSMA-617 RLT at our institution between December 2017 and January 2024 were divided into the following two groups those receiving Lu-177 PSMA-617 RLT plus ARPI (n = 34) and those receiving Lu-177 PSMA-617 RLT alone (n = 70). Patients received 150 to 200 millicuries Lu-177 PSMA-617 RLT in each cycle. PFS and zOS were assessed using Kaplan–Meier analysis and Cox proportional hazard models. **Results:** The combination therapy significantly prolonged median PFS compared to Lu-177 PSMA-617 RLT alone (11 vs. 5.6 months; HR, 0.47; 95% CI, 0.28–0.79; *p* < 0.01). A trend towards improved OS was also observed in the combination group (20.3 vs. 15.9 months; HR, 0.58; 95% CI, 0.33–1.02; *p* = 0.06). Age was a significant predictor of OS (21.2 vs. 12.4 months for younger vs. older patients; *p* < 0.01), while Gleason score and visceral involvement did not significantly impact PFS. The safety profile indicated that adverse effects were generally comparable between the two groups, with no statistically significant differences in the incidence of anemia, neutropenia, thrombocytopenia, nephrotoxicity, or hepatotoxicity. **Conclusions:** This study provides evidence that combining Lu-177 PSMA-617 RLT with ARPIs may significantly improve PFS in mCRPC patients. The potential OS benefit warrants further investigation in larger prospective trials. Age should be considered when making treatment decisions for mCRPC patients.

## 1. Introduction

Prostate cancer continues to be a significant health concern worldwide, with its advanced and metastatic stages causing a substantial number of cancer-related fatalities in men [1]. In the past, managing metastatic prostate cancer has been challenging, particularly when castration-resistant disease (mCRPC) sets in [2]. The occurrence of mCRPC is typically a turning point in the progression of the disease, requiring a change in treatment approaches.

Androgen deprivation therapy (ADT) has long been the foundation of treating metastatic prostate cancer [3]. The therapy effectively suppresses testosterone and disrupts its action, thereby inhibiting the androgen receptor (AR) pathway that drives prostate cancer progression [4]. However, despite initial responses, mCRPC can emerge, whereby the cancer continues to proliferate, even in the presence of castrate levels of testosterone [5]. This underscores the need for innovative therapies to overcome resistance mechanisms. In recent years, significant strides have been made in the treatment of mCRPC. Novel anti-androgen therapies such as abiraterone acetate and enzalutamide display potent inhibition of AR signaling, even in castration-resistant settings. Abiraterone acetate acts specifically to disrupt androgen biosynthesis through CYP17 inhibition [6], while enzalutamide functions as a competitive AR antagonist, blocking androgen binding and subsequent downstream effects [7]. Pivotal trials, including COU-AA-301 and PREVAIL, have demonstrated notable advances in progression-free survival (PFS) and overall survival (OS) with the use of these agents for mCRPC [6,7].

The revolutionary Lutetium-177 (Lu-177) PSMA-617 radioligand therapy (RLT) has emerged as a game-changer in treating mCRPC. This approach involves administering Lu-177 PSMA-617, a radioactive isotope attached to a molecule that selectively binds to prostate-specific membrane antigen (PSMA) [8]. As PSMA is significantly overexpressed on prostate cancer cells, it presents an ideal target for therapy [8,9]. With Lu-177 PSMA-617 RLT, radiation is precisely delivered to tumor cells, causing DNA damage and cell death while minimizing harm to healthy tissue nearby. The landmark VISION trial established the effectiveness of Lu-177 PSMA-617 RLT, demonstrating both PFS and OS advantages over standard-of-care treatments in mCRPC patients who had progressed on prior lines of therapy [10].

The favorable outcomes observed with anti-androgen therapies and Lu-177 PSMA-617 RLT have generated interest in investigating the potential synergies of combination approaches. The idea is to tackle mCRPC from multiple angles, targeting different pathways to address potential resistance mechanisms. Initial studies have shown promising results when combining Lu-177 PSMA-617-based treatments with androgen receptor pathway inhibitors (ARPIs) like abiraterone acetate and enzalutamide, indicating possible enhancements in progression-free survival (PFS) [11]. However, critical questions remain surrounding the optimal selection of patients, sequencing of therapies, and the long-term safety profiles of these combinations. These areas are currently the subject of intense research efforts.

To stay current in this rapidly evolving treatment landscape, oncologists must comprehensively understand the benefits, limitations, and mechanisms of action associated with ARPIs and Lu-177 PSMA-617 RLT. This knowledge is crucial for making personalized, evidence-based treatment decisions that maximize outcomes for patients with advanced prostate cancer. Therefore, our study aims to evaluate the effectiveness of Lu-177 PSMA-617 RLT, either as a standalone treatment or in combination with ARPIs, for patients with mCRPC. We analyze differences in PFS and OS between the treatment groups. Additionally, we investigate how factors like age, Gleason score, prior treatments, visceral involvement, and the line of Lu-177 PSMA-617 RLT may impact treatment outcomes. Our findings have the potential to provide valuable insights to optimize treatment strategies in patients with mCRPC, potentially leading to improved clinical outcomes.

## 2. Materials and Methods

### 2.1. Study Population

This retrospective study included 104 patients diagnosed with mCRPC between December 2017 and January 2024 at Yeditepe University Hospital who received Lu-177 PSMA-617 RLT (Monrol-Eczacıbaşı, Istanbul, Turkiye). Patients were divided into the following two groups to investigate treatment effects: those receiving Lu-177 PSMA-617 RLT plus ARPI (n = 34) and those receiving only Lu-177 PSMA-617 RLT (n = 70) (Figure 1). To be eligible for participation, patients were required to meet the following criteria: radiologically confirmed diagnosis of metastatic prostate cancer; evidence of disease progression despite prior systemic therapy; at least one cycle of Lu-177 PSMA-617 RLT given according to institutional protocols with or without ARPIs; and availability of adequate medical records outlining treatment, imaging, and outcome information.

### 2.2. Administration and Monitoring of Lu-177 PSMA-617 RLT

Patients were given between 150 and 200 millicuries of Lu-177 PSMA-617 in each round. The subsequent cycles took place every 6 weeks. Whole-body scintigraphy and extra single-photon emission computed tomography/computed tomography (SPECT/CT) were conducted at least once 24 to 48 h after the injection to confirm the absorption and retention of Lu-177 PSMA-617 in the tumor tissue.

### 2.3. Data Collection

Demographic information such as age and race/ethnicity; disease characteristics like Gleason score and extent of metastatic disease (visceral vs. non-visceral); prior lines of systemic therapy; treatment details, such as date of first metastasis, date of Lu-177 PSMA-617 RLT initiation, type of ARPIs, and the number of cycles of each therapy; adverse effects; and treatment outcomes were collected from patient medical records.

### 2.4. Assessments

The primary endpoint of this study was PFS, and the secondary endpoints were OS and safety. PFS was calculated as the time from Lu-177 PSMA-617 RLT to disease progression or death in months (whichever occurred first). OS was defined as the time from Lu-177 PSMA-617 RLT initiation until death in months. Response to Lu-177 PSMA-617 RLT was assessed every 2 cycles until disease progression, death, or loss of follow-up for patients who discontinued for any other reason according to the Response Evaluation Criteria in Solid Tumors [12]. [^68^Ga]Ga-68 PSMA Positron Emission Tomography (PET/CT) imaging and the RECIST criteria were used by an expert nuclear radiologist to assess disease progression. A 30% reduction in the SUV max value was defined as regression, and an increase of 30% or more in the max standardized uptake value (SUV) or observation of a new lesion was accepted as progression. Not meeting these criteria was accepted as a stable disease. The safety profile of patients treated with Lu-177 PSMA-617 RLT was systematically evaluated by reviewing medical records and clinical reports. Adverse effects were categorized according to the Common Terminology Criteria for Adverse Events (CTCAE) version 5.0 [13]. Key adverse effects of interest included anemia, neutropenia, thrombocytopenia, nephrotoxicity, and hepatotoxicity. These adverse effects were classified into Grade 1–2 and Grade 3–4 to assess both the incidence and severity.

### 2.5. Statistical Analysis

Statistical methods including descriptive statistics, Kaplan–Meier methods, and a Cox proportional hazards model were used to analyze the data. Statistical analyses were performed using IBM SPSS Statistics for Windows Version 25.0 (Statistical Package for the Social Sciences, IBM Corp., Armonk, NY, USA) A *p*-value < 0.05 was defined as statistically significant.

### 2.6. Ethical Statement

This study was performed in accordance with the principles of the Declaration of Helsinki. The Ethics Committee of Yeditepe University, Istanbul, Turkey, granted approval (date: 15 March 2024/No. E.83321821-805.02.03-379). Informed consent was obtained from all subjects included in the study.

## 3. Results

### 3.1. Demographics and Disease Characteristics

This study investigated the impact of Lu-177 PSMA-617 RLT on PFS and OS in 104 patients diagnosed with mCRPC. The study population had an average age of 64.8 years. Patients were divided into two groups according to treatment. The combination therapy group (n = 34) received ARPIs with Lu-177 PSMA-617 RLT, and the other group received Lu-177 PSMA-617 RLT group (n = 70). The two groups had similar disease and patient characteristics (Table 1). When we examined the correlations of variables, we observed a negative correlation between patient age and first-line treatment choice (*p* = 0.02). There were no significant differences in the first-line and second-line treatments between the groups (*p* = 0.93 and *p* = 0.35, respectively). Most of the patients in both treatment groups received Lu-177 PSMA-617 RLT in the third-line setting (76.5% in the combined group; 68.6% in the 177-Lu PSMA-617 RLT-only group). However, more patients in the combination group had prior exposure to ARPI treatments in the previous lines (94.1% vs. 77.1%; *p* = 0.05). Therefore, we also separately analyzed patients who received prior ARPI. When we examined the correlations of variables, we observed a negative correlation between patient age and first-line treatment choice (*p* = 0.02).

### 3.2. Survival Outcomes

In this study, the analysis of PFS revealed several critical factors influencing outcomes in patients with mCRPC treated with Lu-177 PSMA-617 RLT. Gleason score, prior ARPI treatment, and visceral metastasis did not show a statistically significant impact on PFS. However, a key finding was the significant benefit of combined therapy with Lu-177 PSMA-617 RLT and ARPIs, which resulted in a median PFS of 11.0 months compared to 5.6 months for those receiving Lu-177 PSMA-617 RLT alone (HR, 0.47; CI, 0.28–0.79; *p* < 0.01) (Figure 2). This benefit remained significant even after adjusting for other factors in the multivariate analysis (HR, 0.37; 95% CI, 0.21–0.64; *p* < 0.01) (Table 2).

Additionally, patients who received prior ARPI were further analyzed for differences in PFS based on subsequent treatment with Lu-177 PSMA-617 RLT either combined with ARPI or as monotherapy. It was found that patients who received prior ARPI, then underwent Lu-177 PSMA-617 RLT combined with ARPI upon progression had significantly better PFS compared to those who received Lu-177 PSMA-617 RLT alone (11 months vs. 5.1 months; HR, 0.42; 95% CI: 0.25–0.74; *p* < 0.01) (Figure 3). This also remained significant in multivariate analysis (HR, 0.38; 95% CI, 0.21–0.70; *p* < 0.01) (Table 3).

The OS analysis also highlighted significant predictors of outcomes. Age was a crucial factor, with younger patients (<65 years) demonstrating a markedly improved median OS compared to older patients (HR 0.50; 95% CI 0.30–0.83; *p* < 0.01). While the Gleason score did not significantly affect OS (HR 0.62; 95% CI 0.34–1.10; *p* = 0.10), the presence of visceral metastasis approached statistical significance, suggesting that patients without visceral metastasis had a trend towards better survival (HR 0.65; 95% CI 0.39–1.09; *p* = 0.10). Additionally, prior chemotherapy was associated with a non-significant trend towards improved OS (HR 0.55; 95% CI 0.30–1.00; *p* = 0.05). Although there was a trend towards better survival in the ARPI-naïve group, there was no statistically significant difference in either univariate (27.2 vs. 15.8 months; HR, 0.54; 95% CI, 0.27–1.08; *p* = 0.08) or multivariate analysis (HR, 0.37; 95% CI, 0.10–1.33; *p* = 0.13). Importantly, combined therapy with Lu-177 PSMA-617 RLT and ARPIs showed a trend towards improved OS, with a median of 20.3 months versus 15.9 months for those receiving Lu-177 PSMA-617 RLT alone (HR 0.58; 95% CI, 0.33–1.02; *p* = 0.06) (Figure 4). This trend became significant in multivariate analysis (HR 0.35; 95% CI 0.19–0.67; *p* < 0.01) (Table 4).

Furthermore, the OS data for patients who received prior ARPI were assessed. It was observed that patients who received prior ARPI and subsequently underwent combined treatment with Lu-177 PSMA-617 RLT and ARPI had significantly better OS compared to those who received Lu-177 PSMA-617 RLT alone (24.0 months vs. 14.0 months; HR, 0.37; 95% CI: 0.19–0.70; *p* < 0.01) (Figure 5). This remained significant in multivariate analysis (HR, 0.27; 95% CI, 0.13–0.54; *p* < 0.01) (Table 5).

### 3.3. Safety Profile

Among the Lu-177 PSMA-617 RLT plus ARPI group, 32.3% (11 patients) experienced anemia, with 11.7% (2 patients) having Grade 3–4 anemia. In the Lu-177 PSMA-617 RLT alone group, 24.2% (17 patients) experienced anemia, with 8.5% (6 patients) having Grade 3–4 anemia. Leukopenia was noted in 14.7% (five patients) of patients in the ARPI group, with 5.8% (two patients) having Grade 3–4 leukopenia, while in the Lu-177 PSMA-617 RLT-alone group, 8.5% (six patients) experienced leukopenia, with 4.2% (three patients) having Grade 3–4 leukopenia. Thrombocytopenia occurred in 29.4% (10 patients) of the ARPI group, with 11.7% (4 patients) having Grade 3–4 thrombocytopenia, compared to 27.1% (19 patients) experiencing thrombocytopenia and 10% (7 patients) having Grade 3–4 thrombocytopenia in the Lu-177 PSMA-617 RLT-alone group.

Nephrotoxicity was relatively rare, with 2.9% (one patient) in the ARPI group and 1.4% (one patient) in the Lu-177 PSMA-617 RLT-alone group experiencing any nephrotoxicity. There were no cases of Grade 3–4 nephrotoxicity in either group. Hepatotoxicity was not observed in any patient in either group.

Overall, the safety profile indicates comparable rates of anemia, leukopenia, thrombocytopenia, nephrotoxicity, and hepatotoxicity between the two treatment groups, suggesting that adding ARPI does not significantly increase the risk of these adverse effects. Fisher’s exact test results confirmed that there was no statistically significant difference in the incidence of these adverse effects between the two groups (anemia, *p* = 0.48; leukopenia, *p* = 0.49; thrombocytopenia, *p* = 0.81; nephrotoxicity, *p* = 0.54) (Table 6).

## 4. Discussion

Our study adds valuable insights to the evolving treatment landscape for mCRPC. The results suggest that patients may derive significant benefits when Lu-177 PSMA-617 RLT is combined with ARPIs. The benefit might still exist even if patients stick with the same ARPI despite progression (n = 27) or switch to a different one (n = 5). These findings underscore the potential for synergistic effects when combining Lu-177 PSMA-617 RLT and ARPI treatment modalities and support the evolving treatment paradigm for mCRPC [14]. Our results align with recent Turkish multicenter data, demonstrating improved PFS for combined treatment compared to Lu-177 PSMA-617 RLT monotherapy (11.9 months vs. 7.4 months and 11.0 months vs. 5.6 months, respectively). While OS trends favored combined therapy in our study and the Turkish data (20.3 months vs. 15.9 months and 18.2 months vs. 12.3 months, respectively), the OS differences did not reach statistical significance in either study [15]. This lack of significance might be attributed to the relatively small number of deaths observed in the combination therapy group in our study (17 deaths), potentially resulting in reduced statistical power to detect an actual difference. Further investigations with larger cohorts and a more significant duration of follow-up are crucial to conclusively establish the impact of this combination therapy on OS.

Analysis of prior ARPI treatment revealed important insights. Patients who received prior ARPI, then underwent combined treatment with Lu-177 PSMA-617 RLT and ARPI demonstrated significantly better PFS and OS compared to those who received Lu-177 PSMA-617 RLT alone. Specifically, these patients had a median PFS of 11 months compared to 5 months (HR 0.42; 95% CI: 0.24–0.74; *p* < 0.01) and a median OS of 24.0 months versus 14.0 months (HR 0.37; 95% CI: 0.19–0.70; *p* < 0.01). 

If patients were not previously exposed to ARPI, then received combination therapy, they would naturally have better PFS and OS due to the added therapeutic benefit of ARPI. However, our study’s novel finding is that even patients who had prior ARPI treatment—who might be expected to have diminished responsiveness due to prior exposure—still exhibited significantly improved PFS and OS when treated with the combination of Lu-177 PSMA-617 RLT and ARPI. This suggests that the therapeutic synergy between Lu-177 PSMA-617 RLT and ARPIs can overcome potential resistance mechanisms developed during prior ARPI treatment.

Additionally, it is important to note that patients in the combined treatment group were more frequently exposed to prior ARPI than those in the Lu-177 PSMA-617 RLT-alone group. If the situation were reversed and the Lu-177 PSMA-617 RLT-alone group had more prior ARPI exposure, one could argue that the observed differences in PFS and OS might be attributed to the effects of prior ARPI exposure. However, the fact that the combined treatment group had more prior exposure yet still showed superior outcomes reinforces the efficacy of the combination therapy.

Several hypotheses exist as to why dual treatment with Lu-177 PSMA-617 RLT and ARPI may be advantageous. Targeting heterogeneous tumor populations in advanced prostate cancer is crucial, as these tumors often have some cells retaining AR dependence and others developing AR-independent resistance mechanisms [16,17]. Additionally, AR pathway mutations usually drive ARPI resistance in mCRPC [18], and Lu-177 PSMA-617 RLT AR-independent action may help to counteract this resistance.

While age, Gleason score, and visceral involvement did not display a statistically significant impact on PFS outcomes in our cohort, these factors remain crucial considerations in personalized treatment planning. The observed correlation between older patient age and less aggressive treatment choices, both in initial and subsequent treatment lines, highlights the complex decision-making process for this population. Balancing potential benefits against tolerability, particularly with chemotherapy, is vital to providing optimal care. This is especially important as studies reveal that age may not be a contraindication for Lu-177 PSMA-617 RLT, with elderly patients potentially deriving similar benefits to those experienced by younger counterparts [19].

Identifying age as a significant predictor of OS aligns with established clinical understanding. Advanced age often corresponds with increased frailty and potential ineligibility for aggressive subsequent treatment regimens, possibly contributing to decreased OS. Older patients might be less likely to receive chemotherapy due to concerns regarding their overall health and ability to tolerate it. This limited access to treatment options can explain the survival difference between age groups.

The timing of Lu-177 PSMA-617 RLT within a patient’s treatment sequence emerges as another significant variable. Our data echo the real-world trend of reserving Lu-177 PSMA-617 RLT for later lines of treatment [20]. Similarly, the landmark VISION trial demonstrated the efficacy of Lu-177 PSMA-617 RLT, even in patients who had progressed under multiple prior treatments [10]. As Lu-177 PSMA-617 RLT gains recognition and approval, understanding its optimal placement in the treatment algorithm becomes an urgent area of active research. Recent studies such as the PSMAfore trial (NCT04689828) have demonstrated that Lu-177 PSMA-617 RLT provides benefits in earlier lines in terms of PFS and the objective response rate for taxane-naive patients with mCRPC [21].

The evaluation of the safety profile in our study revealed that the adverse effects of Lu-177 PSMA-617 RLT are generally comparable between the two treatment groups, with similar rates of anemia, neutropenia, and thrombocytopenia. Our findings are consistent with those from the TheraP trial, which reported fewer Grade 3–4 toxicities with Lu-177 PSMA-617 RLT compared to cabazitaxel, highlighting the low-toxicity profile of Lu-177 PSMA-617 RLT [22]. However, due to the retrospective design of our study and potential gaps in the recording of side effect profiles in patient files, it is acknowledged that some side effects based on patient complaints may not have been accurately evaluated. Therefore, prospective studies with larger cohorts are necessary to determine if combination therapy is safe.

This study reinforces the importance of multidisciplinary collaboration in addressing the challenges of mCRPC. The complex interaction between disease, patient characteristics, and expanding treatment options underscores the need for personalized treatment strategies. Oncologists must understand novel therapies like Lu-177 PSMA-617 RLT and ARPI and the evolving evidence supporting their use in combination regimens.

Certain limitations of our study should be acknowledged. The retrospective design introduces inherent biases. Additionally, the relatively small sample size, relatively short follow-up time, and single-center focus may limit the generalizability of our findings. Larger prospective studies are needed to further validate the potential benefits of combining Lu-177 PSMA-617 RLT with ARPI and to refine patient selection criteria for optimal outcomes.

Despite these limitations, our work underscores the rapid progress in mCRPC treatment. The combined approach of Lu-177 PSMA-617 RLT and ARPI represents a promising strategy to improve outcomes in this challenging patient population. Ongoing research is essential to determine the optimal sequencing of therapies, explore potential biomarkers for treatment response, and establish long-term safety profiles of these combinations. Further investigations will undoubtedly refine treatment paradigms and lead to even greater personalization of care for advanced prostate cancer.

## Figures and Tables

**Figure 1 jcm-13-04585-f001:**
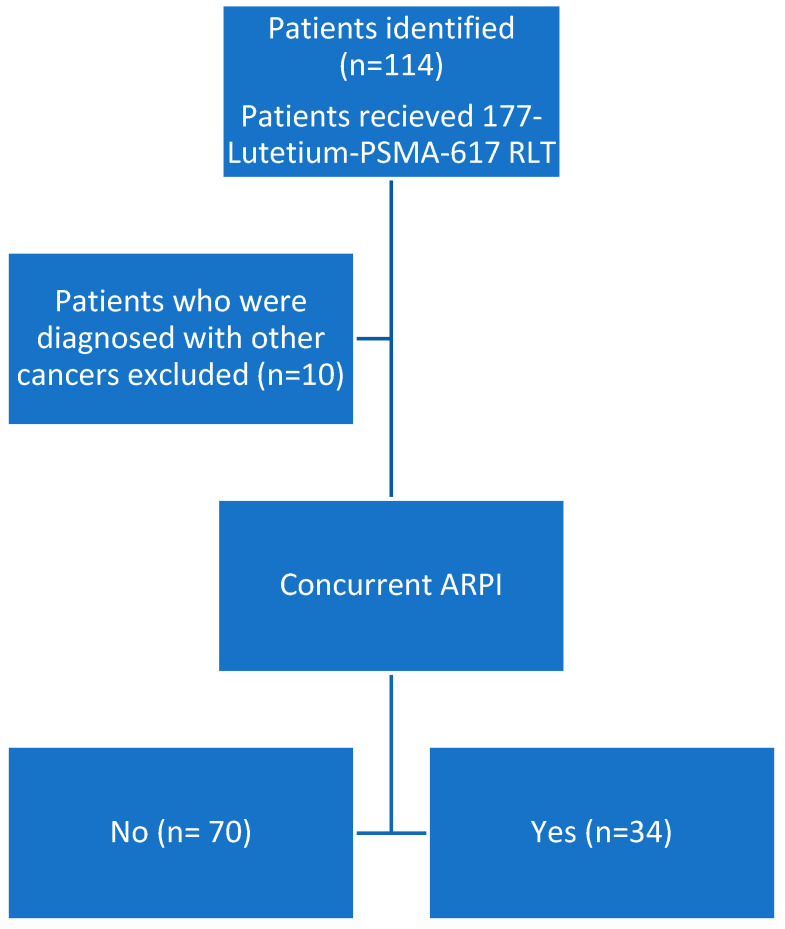
Flow chart describing patients receiving 177-Lutetium-PSMA-617. Abbreviations: ARPI, androgen receptor pathway inhibitor; RLT, radioligand therapy.

**Figure 2 jcm-13-04585-f002:**
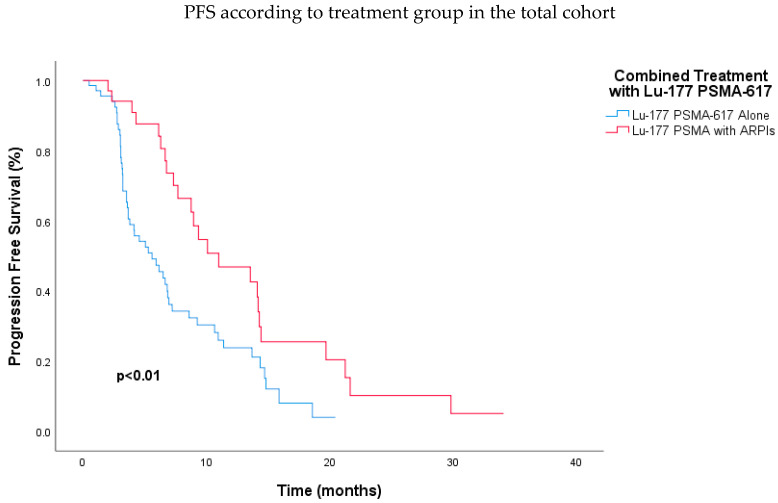
Kaplan–Meier analysis demonstrating significantly longer progression-free survival for patients receiving combined Lu-177 PSMA-617 RLT and androgen receptor pathway inhibitors compared to those receiving Lu-177 PSMA-617 RLT alone (*p* < 0.01). Abbreviations: ARPI, androgen receptor pathway inhibitor; Lu-177-PSMA, lutetium-177 PSMA.

**Figure 3 jcm-13-04585-f003:**
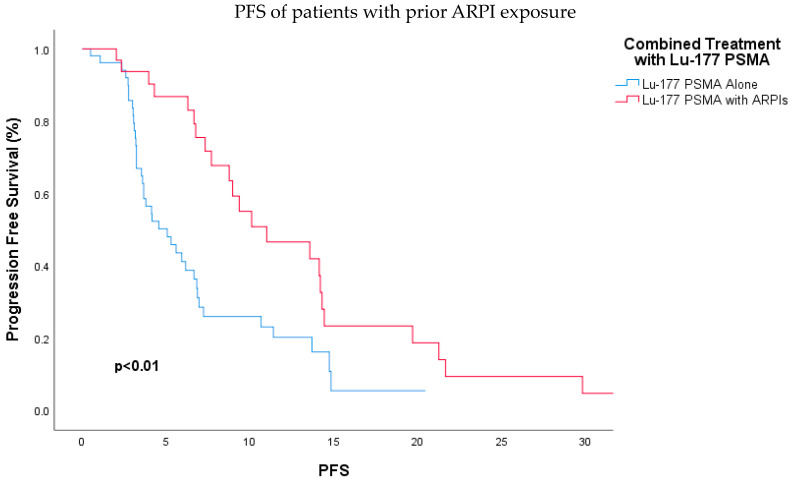
Kaplan–Meier analysis demonstrating significantly longer progression-free survival for patients receiving combined Lu-177 PSMA-617 RLT and androgen receptor pathway inhibitors compared to those receiving Lu-177 PSMA-617 RLT alone among those previously treated with an ARPI (*p* < 0.01). Abbreviations: ARPI, androgen receptor pathway inhibitor; Lu-177-PSMA, lutetium-177 PSMA.

**Figure 4 jcm-13-04585-f004:**
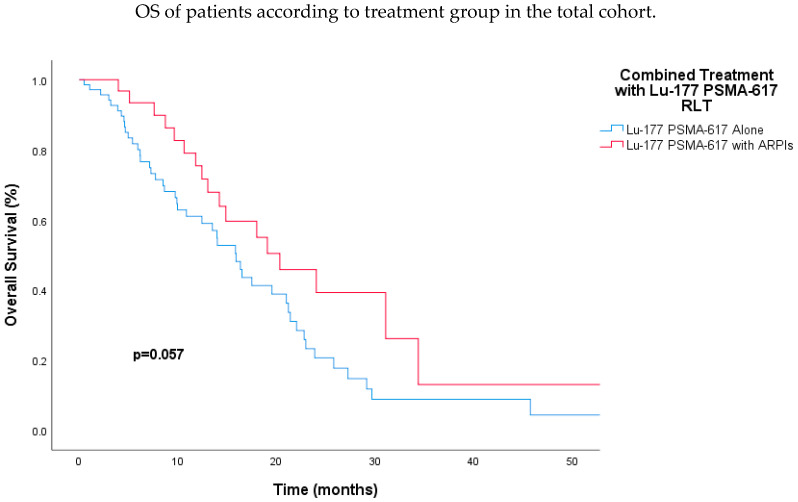
Kaplan–Meier analysis demonstrating a trend toward better overall survival for patients receiving combined Lu-177 PSMA-617 RLT and androgen receptor pathway inhibitor therapy versus those receiving Lu-177 PSMA-617 RLT alone. Abbreviations: ARPI, androgen receptor pathway inhibitor; Lu-177-PSMA, lutetium-177-PSMA.

**Figure 5 jcm-13-04585-f005:**
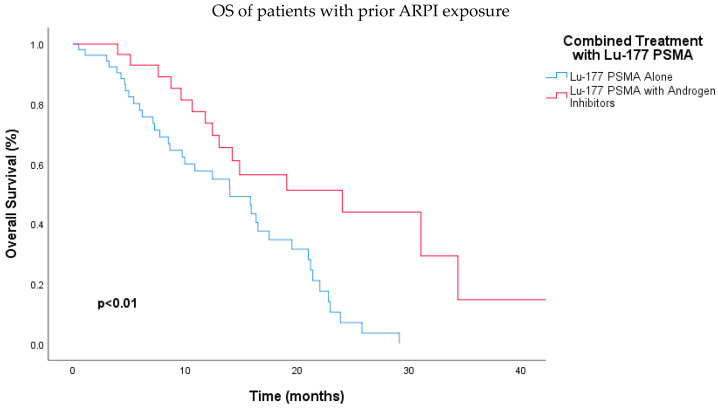
Kaplan–Meier analysis demonstrating significantly longer overall survival for patients receiving combined Lu-177 PSMA-617 RLT and androgen receptor pathway inhibitors compared to those receiving Lu-177 PSMA-617 RLT alone among those who were previously treated with an ARPI (*p* < 0.01). Abbreviations: ARPI, androgen receptor pathway inhibitor; Lu-177-PSMA, lutetium-177 PSMA.

**Table 1 jcm-13-04585-t001:** Distribution of the clinical characteristics of the patients.

Characteristic	Total Cohort, n (%)	With ARPI, n (%)	Without ARPI, n (%)	*p*-Value
Total Patients	104	34 (32.7)	70 (67.3)	
Age				
Mean	64.89 (44–85)	66.41 (50–85)	64.16 (44–81)	
Risk categories based on Gleason grading				0.94
Low	0	0 (0)	0 (0)	
Intermediate	28 (26.9)	9 (26.5)	19 (27.1)	
High/very high	76 (73.1)	34 (73.5)	51 (72.9)	
First-line Treatment				0.93
Docetaxel	76 (73.1)	24 (70.6)	52 (74.3)	
ADT alone	11 (10.6)	3 (8.8)	8 (11.4)	
Enzalutamide	13 (12.5)	5 (14.7)	8 (11.4)	
Abiraterone	4 (3.8)	2 (5.9)	2 (2.9)	
Second-line Treatment				0.35
Docetaxel	8 (7.7)	2 (5.9)	6 (8.6)	
Enzalutamide	42 (40.4)	14 (41.2)	28 (40.0)	
Abiraterone	23 (22.1)	10 (29.4)	13 (18.6)	
Lu-177 PSMA-617 RLT	30 (28.8)	8 (23.5)	22 (31.4)	
Cabazitaxel	1 (1)	0 (0)	1 (1.4)	
Lu-177 PSMA-617 RLT line				0.49
2	30 (28.8)	8 (23.5)	22 (31.4)	
≥3	74 (71.2)	26 (76.5)	48 (68.6)	
Metastasis site of prior Lu-177 PSMA-617 RLT				0.37
Only Bone	69 (66.3)	25 (73.5)	44 (62.9)	
Visceral or visceral plus bone	35 (33.7)	9 (26.5)	26 (37.1)	
Lu-177 PSMA RLT cycle, median (min–max)	4 (1–8)	4 (1–7)	3 (1–8)	
Any Prior Chemotherapy				0.27
Yes	86 (82.7)	26 (76.5)	60 (85.7)	
No	18 (17.3)	8 (23.5)	10 (14.3)	
Prior ARPI				0.05
Yes	86 (82.7)	32 (94.1)	54 (77.1)	
No	18 (17.3)	2 (5.9)	16 (22.9)	

Abbreviations: ARPI, androgen receptor pathway inhibitor; ADT, androgen deprivation treatment; Lu-177 PSMA-617 RLT, lutetium-177 PSMA-617 radioligand therapy.

**Table 2 jcm-13-04585-t002:** Univariate and multivariate analyses of factors associated with PFS in patients with mCRPC treated with Lu-177 PSMA-617 RLT.

	Univariate Analysis			Multivariate Analysis	
	PFS (Months)	*p* Value	HR (95% CI)	*p* Value	HR (95% CI)
Age					
<65	9.2	0.33	0.79 (0.50–1.26)	0.09	0.64 (0.39–1.06)
≥65	6.6				
Gleason Group					
Intermediate	14.7	0.32	0.77 (0.46–1.28)	0.57	0.84 (0.46–1.53)
High/very high	11.0				
Visceral Metastasis					
No	7.7	0.32	0.79 (0.49–1.26)	0.45	0.83 (0.50–1.35)
Yes	4.3				
Prior Chemotherapy					
No	6.9	0.78	0.91 (0.49–1.71)	0.44	0.71 (0.29–1.69)
Yes	6.6				
Lu-177 PSMA-617 RLT line					
2	8.6	0.61	0.87 (0.52–1.45)	0.16	0.67 (0.38–1.18)
≥3	6.8				
Prior ARPI					
Yes	6.8	0.53	0.82 (0.45–1.51)	0.052	0.38 (0.14–1.01)
No	9.2				
Combined Therapy					
Yes	11.0	<0.01	0.47 (0.28–0.79)	<0.01	0.37 (0.21–0.64)
No	5.6				

Abbreviations: PFS: progression-free survival; Lu-177 PSMA-617 RLT: lutetium-177 PSMA-617; HR: hazard ratio; mCRPC: metastatic castration-resistant prostate cancer.

**Table 3 jcm-13-04585-t003:** Univariate and multivariate analyses of factors associated with PFS for patients who were previously exposed to ARPI.

	Univariate Analysis			Multivariate Analysis	
	PFS (Months)	*p* Value	HR (95% CI)	*p* Value	HR (95% CI)
Age					
<65	6.83	0.48	0.83 (0.50–1.39)	0.11	0.63 (0.36–1.11)
≥65	6.86				
Gleason Group					
Intermediate	7.7	0.68	0.88 (0.49–1.58)	0.68	0.87 (0.47–1.64)
High/very high	6.7				
Visceral Metastasis					
Yes	4.3	0.21	0.72 (0.43–1.21)	0.52	0.83 (0.49–1.42)
No	7.3				
Prior Chemotherapy					
Yes	6.8	0.97	0.98 (0.49–2.02)	0.58	0.61 (0.10–3.52)
No	7.2				
Lu-177 PSMA-617 RLT line					
2	7.3	0.99	0.99 (0.47–2.1)	0.59	0.61 (0.99–3.74)
≥3	6.8				
Combined Therapy					
Yes	11.0	<0.01	0.43 (0.25–0.74)	<0.01	0.39 (0.21–0.70)
No	5.1				

Abbreviations: PFS: progression-free survival; Lu-177 PSMA-617 RLT: lutetium-177 PSMA-617; HR: hazard ratio; mCRPC: metastatic castration-resistant prostate cancer.

**Table 4 jcm-13-04585-t004:** Univariate and multivariate analyses of factors associated with OS in patients with mCRPC treated with Lu-177 PSMA-617 RLT.

	Univariate Analysis			Multivariate Analysis	
	OS (Months)	*p* Value	HR (95% CI)	*p* Value	HR (95% CI)
Age					
<65	21.2	<0.01	0.50 (0.30–0.83)	<0.01	0.37 (0.21–0.67)
≥65	12.4				
Gleason Group					
Intermediate	24.0	0.10	0.62 (0.34–1.10)	0.54	0.82 (0.43–1.56)
High/very high	16.3				
Visceral Metastasis					
No	19.0	0.10	0.65 (0.39–1.09)	0.08	0.62 (0.37–1.06)
Yes	13.5				
Prior Chemotherapy					
No	11.8	0.05	0.55 (0.30–1.00)	0.81	0.90 (0.39–2.1)
Yes	17.5				
Lu-177 PSMA-617 RLT line					
2	20.3	0.36	0.77 (0.44–1.34)	0.89	0.93 (0.32–2.66)
≥3	15.8				
Prior ARPI					
Yes	15.8	0.08	0.54 (0.27–1.08)	0.13	0.37 (0.10–1.33)
No	27.2				
Combined Therapy					
Yes	20.3	0.06	0.58 (0.33–1.02)	<0.01	0.35 (0.19–0.67)
No	15.9				

Abbreviations: PFS: progression-free survival; Lu-177 PSMA-617 RLT: lutetium-177 PSMA-617; HR: hazard ratio; mCRPC: metastatic castration-resistant prostate cancer.

**Table 5 jcm-13-04585-t005:** Univariate and multivariate analyses of factors associated with OS for patients who were previously exposed to ARPI.

	Univariate Analysis			Multivariate Analysis	
	OS (Months)	*p* Value	HR (95% CI)	*p* Value	HR (95% CI)
Age					
<65	17.5	0.06	0.59 (0.34–1.02)	0.02	0.36 (0.19–0.69)
≥65	11.8				
Gleason Group					
Intermediate	14.2	0.66	0.87 (0.46–1.63)	0.89	0.89 (0.44–1.83)
High/very high	16.3				
Visceral Metastasis					
Yes	14.0	0.27	0.73 (0.42–1.28)	0.30	0.73 (0.40–1.33)
No	16.5				
Prior Chemotherapy					
Yes	15.8	0.52	0.80 (0.40–1.59)	0.22	0.46 (0.13–1.60)
No	16.5				
Lu-177 PSMA-617 RLT line					
2	16.5	0.48	0.77 (0.37–1.59)	0.22	0.48 (0.15–1.56)
≥3	15.8				
Combined Therapy					
Yes	24.0	<0.01	0.37 (0.19–0.70)	<0.01	0.27 (0.13–0.54)
No	14.0				

Abbreviations: PFS: progression-free survival; Lu-177 PSMA-617 RLT: lutetium-177 PSMA-617; HR: hazard ratio; mCRPC: metastatic castration-resistant prostate cancer.

**Table 6 jcm-13-04585-t006:** Comparison of safety profile for key adverse effects in patients treated with Lu-177 PSMA-617 RLT.

Adverse Effect	Lu-177 PSMA-617 RLT Plus ARPI (n = 34)		Lu-177 PSMA-617 RLT Alone (n = 70)		*p* Value
	Any Event	Grade 3–4	Any Event	Grade 3–4	
Anemia	11 (32.3)	2 (11.7)	17 (24.2)	6 (8.5)	0.48
Leukopenia	5 (14.7)	2 (5.8)	6 (8.5)	3 (4.2)	0.49
Thrombocytopenia	10 (29.4)	4 (11.7)	19 (27.1)	7 (10.0)	0.81
Nephrotoxicity *	1 (2.9)	0 (0)	1 (1.4)	0 (0)	NA
Hepatotoxicity ^†^	0 (0)	0 (0)	0 (0)	0 (0)	NA

Abbreviations: ARPI; androgen receptor pathway inhibitor. * Nephrotoxicity was assessed based on creatinine elevation. ^†^ Hepatotoxicity was assessed based on alanine transaminase elevation.

## Data Availability

The datasets generated and/or analyzed during the current study are available from the corresponding author upon reasonable request.

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
