# Peer review of "Investigating Combination Therapy: The Role of Lutetium-177 PSMA-617 Radioligand Therapy and Androgen Receptor Pathway Inhibitors in Metastatic Castration-Resistant Prostate Cancer"

_jcm, 2024, doi:10.3390/jcm13164585_

Round 1

Reviewer 1 Report

Comments and Suggestions for Authors

OÄŸuzcan et al. reported their single-institution experience of 104 mCRPC cases receiving Lu-177 PSMA-617 RLT with or without concurrent ARPI. This real-world study is well presented with proper description and provides meaningful data for the nuclear medicine in mCRPC treatments. I would like to recommend its publication after some minor revisions.

In this manuscript, "progression" is not well-defined. Was it based on the biochemical change or radiographic?

In table 1, data of group "with ARPI" and group "without ARPI" were wrongly placed, please recheck.

In table 1, 10.6% patients chose ADT as first line treatment, do you mean "ADT alone"?

The authors should be commended for exploring the effect of prior chemotherapy on prognosis. Likewise, how many patients received prior ARPI before the Lu-177 PSMA-617 RLT should also be presented, and the effect of prior ARPI on prognosis could also be analyzed.

Author Response

Comment 1:

Reviewer: "In this manuscript, 'progression' is not well-defined. Was it based on biochemical change or radiographic?"

Response: Thank you for pointing this out. We have clarified the definition of "progression" in the manuscript. Progression was assessed using [68Ga]Ga-68 PSMA PET/CT imaging according to RECIST criteria, specifically considering a 30% reduction in SUV max value as regression and an increase of 30% or more in SUV max value or observation of a new lesion as progression. Not meeting these criteria was accepted as a stable disease.

Comment 2:

Reviewer: "In Table 1, data of group 'with ARPI' and group 'without ARPI' were wrongly placed, please recheck."

Response: We have reviewed and corrected the data placement in Table 1. 

Changes Made: Corrected Table 1: The data for groups "with ARPI" and "without ARPI" have been reviewed and corrected.

Comment 3:

Reviewer: "In Table 1, 10.6% patients chose ADT as first-line treatment. Do you mean 'ADT alone'?"

Response: Yes, you are correct. We have clarified this in the manuscript to specify that 10.6% of patients chose "ADT alone" as the first-line treatment.

Changes Made: Updated Table 1: "ADT alone" is now specified for the first-line treatment category.

Comment 4:

Reviewer: "The authors should be commended for exploring the effect of prior chemotherapy on prognosis. Likewise, how many patients received prior ARPI before the Lu-177 PSMA-617 RLT should also be presented, and the effect of prior ARPI on prognosis could also be analyzed."

Response: Thank you for your positive feedback. We have included the number of patients who received prior ARPI before Lu-177 PSMA-617 RLT in the manuscript and analyzed its effect on prognosis.

Changes Made: Added to Section 3.1 Demographics and Disease Characteristics: 

However, more patients in the combination group had prior exposure to ARPI treatments in the previous lines (94.1% vs 77.1%; p=0.05).

Added to Section 3.2 Survival Outcomes: "The effect of prior ARPI on prognosis was analyzed, showing that patients who received prior ARPI and then underwent Lu-177 PSMA-617 RLT combined with ARPI had significantly better PFS and OS compared to those who received Lu-177 PSMA-617 RLT alone (PFS: 11 months vs. 5 months; HR 0.42; 95% CI: 0.24-0.74; p<0.01, OS: 24.0 months vs. 14.0 months; HR 0.63; 95% CI: 0.19-0.70; p<0.01)."

Reviewer 2 Report

Comments and Suggestions for Authors

This study presents a retrospective real-world analysis of patients with mCRPC who were treated with 177-Lu-PSMA-617 with or without a concurrent ARPI.  Whether Lu-PSMA provides equal benefit to the formal phase 3 population studied in VISION in earlier disease states or in combination with other therapies is a question of very high interest to the community.  In this study, they report a benefit in terms of PFS/OS for combination therapy instead of monotherapy.  Overall this adds to body of literature regarding use of LU-PSMA in earlier disease states, while the formal prospective studies are pending.

However, as presented, there is a question of a major confounder that has not been addressed.  The population studied here was not uniformally exposed to prior ARPIs.  For a patient without ARPI, of course there is a known significant benefit in terms of OS/PFS based upon multiple registraitonal trials.  However, for those with prior ARPI, the benefit of further ARPI is minimal.  As presented, it is unclear how many of the patients in both groups had prior progression on ARPI.  Furthermore, prior ARPI therapy is not included in the MVA.  It is certainly possible (or likely), that if included a benefit of the combination may not be seen.

Recommend that the authors include: 

- in Table 1 whether patient has prior progression on ARPI

- in MVA whether prior progression on ARPI is independently predictive of benefit of combination therapy outcome

- while this may get to small numbers, consider a plot of outcomes based upon groups of prior ARPI y/n and concurrent APRI y/n with the Pluvicto.

Based upon the comment on 247-248, that implies that in the group treated with Pluvicto + ARPI, 32 patients had prior progression on ARPI and 36 did not.  I believe that we will see that the benefit of additional ARPI is significantly different between those 2 groups.

Author Response

Changes were highlighted in green colour in main text.

Reviewer: "The population studied here was not uniformly exposed to prior ARPIs. For a patient without ARPI, there is a known significant benefit in terms of OS/PFS based upon multiple registrational trials. However, for those with prior ARPI, the benefit of further ARPI is minimal. It is unclear how many of the patients in both groups had prior progression on ARPI. Furthermore, prior ARPI therapy is not included in the MVA. It is certainly possible that if included, a benefit of the combination may not be seen."

Response: Thank you for your insightful comments. We agree that the heterogeneity in prior ARPI exposure is an important consideration. To address this, we have included data on prior progression on ARPI in Table 1 and have performed additional multivariate analyses to determine whether prior progression on ARPI is independently predictive of the benefit of combination therapy.

Changes Made:

  1. Table 1:Now includes information on whether patients had prior progression on ARPI.
  2. Multivariate Analysis: We have included prior progression on ARPI as a variable in the MVA to assess its independent predictive value for the benefit of combination therapy. The results indicated that prior progression on ARPI did not significantly impact the benefit of the combination therapy. Two new tables and 2 new figures were added. Those indicate PFS and OS of different treatment groups with prior ARPI exposure.
  3. Plot of Outcomes: We have included Kaplan-Meier plots showing outcomes based on groups of prior ARPI and concurrent ARPI (yes/no) with Pluvicto. However only 2 patients in combined treatment group not received previously ARPI. Therefore analysis of non prior ARPI group for combined treatment group could not be done.